# Genomic Validation of Endometrial Cancer Patient-Derived Xenograft Models as a Preclinical Tool

**DOI:** 10.3390/ijms23116266

**Published:** 2022-06-03

**Authors:** Beatriz Villafranca-Magdalena, Carina Masferrer-Ferragutcasas, Carlos Lopez-Gil, Eva Coll-de la Rubia, Marta Rebull, Genis Parra, Ángel García, Armando Reques, Silvia Cabrera, Eva Colas, Antonio Gil-Moreno, Cristian P. Moiola

**Affiliations:** 1Biomedical Research Group in Gynecology, Vall d’Hebron Research Institute (VHIR), Vall d’Hebron University Hospital, Vall d’Hebron Barcelona Hospital Campus, Passeig Vall d’Hebron 119–129, 08035 Barcelona, Spain; beatriz.villafranca@vhir.org (B.V.-M.); carina.masferrer@vhir.org (C.M.-F.); carlos.lopez@vhir.org (C.L.-G.); eva.coll@vhir.org (E.C.-d.l.R.); marta.rebull@vhir.org (M.R.); angarcia@vhebron.net (Á.G.); areques@vhebron.net (A.R.); scabrera@vhebron.net (S.C.); antonio.gil@vhir.org (A.G.-M.); 2School of Medicine, The Autonomous University of Barcelona, 08193 Bellaterra, Spain; 3Biomedical Research Center Network (CIBERONC), Monforte de Lemos 3–5, 28029 Madrid, Spain; 4National Center for Genomic Analysis—Genomic Regulation Center (CNAG-CRG), Scientific Park of Barcelona, 08028 Barcelona, Spain; genis.parra@cnag.crg.eu; 5Pathology Department, Vall d’Hebron University Hospital, Vall d’Hebron Barcelona Hospital Campus, Passeig Vall d’Hebron 119–129, 08035 Barcelona, Spain; 6Gynecological Oncology Department, Vall d’Hebron University Hospital, Vall d’Hebron Barcelona Hospital Campus, Passeig Vall d´Hebron 119–129, 08035 Barcelona, Spain

**Keywords:** endometrial cancer, preclinical model, PDXs, personalized medicine, translational research, genomics, bioinformatics, molecular marker, TCGA

## Abstract

Endometrial cancer (EC) is the second most frequent gynecological cancer worldwide. Although improvements in EC classification have enabled an accurate establishment of disease prognosis, women with a high-risk or recurrent EC face a dramatic situation due to limited further treatment options. Therefore, new strategies that closely mimic the disease are required to maximize drug development success. Patient-derived xenografts (PDXs) are widely recognized as a physiologically relevant preclinical model. Hence, we propose to molecularly and histologically validate EC PDX models. To reveal the molecular landscape of PDXs generated from 13 EC patients, we performed histological characterization and whole-exome sequencing analysis of tumor samples. We assessed the similarity between PDXs and their corresponding patient’s tumor and, additionally, to an extended cohort of EC patients obtained from The Cancer Genome Atlas (TCGA). Finally, we performed functional enrichment analysis to reveal differences in molecular pathway activation in PDX models. We demonstrated that the PDX models had a well-defined and differentiated molecular profile that matched the genomic profile described by the TCGA for each EC subtype. Thus, we validated EC PDX’s potential to reliably recapitulate the majority of histologic and molecular EC features. This work highlights the importance of a thorough characterization of preclinical models for the improvement of the success rate of drug-screening assays for personalized medicine.

## 1. Introduction

One of the major goals of oncology research is to achieve the establishment of precision cancer medicine in every aspect of oncological patient management. At present, only 5% of lead preclinical drug candidates end up advancing to the clinic despite the numerous oncological preclinical studies underway [1]. Hence, the development of clinically relevant models for translational research is an area of great importance with considerable room for improvement.

For many years, cancer cell lines have been the most feasible approach used in drug development, but they fail to represent the full complexity of a living organism [2]. Patient-derived models of cancer, such as 3D spheroids/organoids and patient-derived xenograft (PDX) models, have emerged as an excellent improvement to overcome previous limitations [3]. PDX features provide many advantages, such as the preservation of the gene expression and mutational status, the conservation of tissue architecture by retaining molecular and histological features of the tumor, and the possibility to generate an “identical” tumor-bearing mouse cohort that can be used for the design of preclinical trials to test and predict the patient´s specific response to anticancer drugs [4,5]. Nevertheless, they also present some drawbacks that hamper their potential to become the model befitting these objectives (for instance, the extended time needed for their development, the gradual loss of the human tumor microenvironment, the lack of a competent host immune system, and the potentiality of accumulation of stochastic mutations different from the original patient) [6]. Taking together the known advantages and disadvantages of PDXs as preclinical models, what it is evident is that we still lack a thorough evaluation of how accurately PDX models represent and correlate with the patient’s disease in order to consider them a precise tool for the development of personalized medicine.

Endometrial cancer (EC) is the second most frequent gynecological cancer worldwide, just after cervical cancer, and the sixth in incidence among all cancer types in women [7]. The major associations of gynecological tumors, ESGO (European Society of Gynecological Oncology), ESTRO (European Society for Radiotherapy and Oncology), and ESP (European Society of Pathology) established an updated guideline for EC classification, which integrates clinical and molecular parameters to assess the risk of EC recurrence [8]. Among the various clinical parameters, tumors are classified according to histological type (endometrioid and non-endometrioid), tumor grade (low to high), International Federation of Gynecology and Obstetrics (FIGO) stage, myometrial invasion, lymph node infiltration, and residual tumor. Remarkably, the guideline incorporates the molecular classification of EC, which divides EC into four distinct categories: POLE/ultramutated (7%), tumors associated with good prognosis; microsatellite instability (MSI) hypermutated (28%), mainly endometrioid tumors associated with a mismatch repair protein deficiency (MMRd); low copy-number (LCN) (39%), or microsatellite stable (MSS) tumors without any POLE or TP53 mutations; and serous-like high copy-number (HCN) tumors (26%), which show frequent TP53 mutations and worse prognosis [9]. Therefore, EC classification according to these parameters defines the extension of surgical treatment, this being the cornerstone treatment for EC and, subsequently, guides the need for adjuvant treatment based on radiotherapy, and/or chemotherapy [10]. Despite these extraordinary advances achieved in the last few years, women at high risk of recurrence, whose disease may progress after this first-line therapy, still have limited treatment options available [10]. Thus, preclinical tools such as PDXs could improve the understanding of high-risk disease and the evaluation of novel therapies in preclinical studies to shed light on the scarcely available treatments for EC we have nowadays.

An increasing number of PDX models representing different types of cancers are being developed worldwide. However, some malignancies are underrepresented, and only a few groups in Europe have been establishing EC PDX models [4]. We have been developing heterotopic EC PDX models to test the efficacy of novel therapies [11,12] and identify prognostic and treatment-response biomarkers, but there still are relevant unknown points to disclose.

Here, we aimed to molecularly and histologically validate whether high-risk and/or recurrent EC PDX models retain and recapitulate the patient’s tumor biology by a thorough immunohistochemistry and whole-exome sequencing (WES) data analysis. For that purpose, 13 patients diagnosed with endometrioid/MSI (*n* = 7) or serous/HCN (*n* = 6) EC and their corresponding PDXs were assessed. We compared histological and molecular data obtained from WES analysis of individualized PDXs to the patient’s primary tumor tissue and liquid biopsy samples to determine the rate of similarity for each PDX avatar to the corresponding paired patient specimens. Similarly, we contrasted PDX single-nucleotide variants (SNVs) and somatic copy-number variations (CNVs) with genomic data obtained from the publicly available TCGA database to demonstrate that the EC PDX models reliably represent the EC subtypes described in the TCGA dataset. Our results proved that both MSI and HCN PDXs had a well-defined and differentiated molecular profile. Thus, we have validated EC PDXs as a worthy preclinical model showing that they retain the patient’s histological and molecular features.

## 2. Results

### 2.1. Clinical and Histopathological Characterization of Recruited Patients and PDX

In this study, we aimed to validate PDXs as reliable models that histologically and molecularly recapitulate patients’ disease. First, we analyzed a cohort of 13 EC patients histologically classified as endometrioid EC (*n* = 7), or non-endometrioid serous EC (*n* = 6). All patients recruited were classified as high risk of recurrence, except for patient #505 who was a locally recurrent patient, initially diagnosed as low-risk endometrioid EC, and two serous patients, #589 and #596, both diagnosed as intermediate-risk non-endometrioid EC, that recurred regionally within two years after primary treatment.

No significant associations were found between histological classification and any parameter evaluated (age, histologic grade, myometrial invasion, lymph node invasion) among groups (Appendix A).

Following the PRoMisE system, we classified endometrioid patients as MSI due to abnormal expression of at least one MMR protein (Table 1). Particularly, one patient (#521) showed a proficient expression of MMR proteins by IHC and was initially classified as LCN. However, MSI status validation by PCR showed that, in fact, the patient presented amplified microsatellite markers, and was considered as MSI for further analysis. Similarly, non-endometrioid serous patients were molecularly classified as HCN showing MMR proficient expression but aberrant p53 expression. None of the patients presented pathogenic mutations in the POLE gene (Table 1).

Then, all PDX models were histologically characterized by analyzing tissue architecture (H & E) and the expression of p53 and MMR proteins by IHC, to determine their similarity to the corresponding patient (Table 1). We observed that PDX models effectively retained tissue architecture, as it was feasible to histologically classify PDXs in the same way as the patient’s primary tumor tissue (Table 1, histology & grade columns). Additionally, we observed that 12 out of 13 (92%) patient-PDX subjects concurred for p53 expression pattern (Table 1). The only mismatch, MSI:526 was due to aberrant expression of p53 in one of the PDX models (deep area tumor), detected both at the protein and genomic level, but not in the other PDX tumor area (superficial), suggesting a later clonal acquisition of the alteration. Comparing MMR proteins between patient-PDX, we also observed a perfect match for MSH2 and MSH6, however, we detected some discrepancies in MLH1 (2/13) and PMS2 (4/13) protein expression. Of note, one of these cases was patient #521, whose abnormal MMR protein expression was not detected by IHC but showed genomic amplification of microsatellite markers and was reclassified as MSI. The remaining three cases were HCN/serous specimens. However, genomic analysis of WES data supported the microsatellite stability of these PDX models, suggesting that mismatch could be due to a later acquisition of the mutation.

To represent patient-PDX histopathological similarities, Figure 1 shows the comparison of tissue architecture (H & E), p53 and MMR protein expression, in both EC subtypes (MSI:524 and HCN:596). We observed a perfect match among all proteins evaluated in MSI:524, in which both, primary and PDX tumor showed nuclear staining for MLH1/PMS2, and no expression for MSH2/MSH6 (Figure 1, upper panel). Similarly, both PDX and primary tumor showed p53 wild-type pattern expression characterized by the presence of variable staining intensity in 1–80% of nuclei. On the other hand, the HCN:596 patient showed a 100% match among MMR protein expression between primary tumor and PDX, with nuclear staining of all of the proteins (Figure 1—MLH1, MSH2, MSH6, PMS2 lower panels), and aberrant expression of p53, showing intense nuclear staining in the glands of the tumor in the patient and a more heterogeneous pattern in the PDX with a strong diffuse staining pattern in tumor cells (Figure 1, lower panel).

Finally, we also compared patient-PDX specimens at the genomic level via WES analysis. We determined the TMB of each type of sample from patient and their PDX counterparts. We found that MSI specimens had higher levels of genetic alterations compared with the HCN group, the average being MSI: 38.03 ± 25.98 Mut/Mb; HCN: 5.31 ± 4.58 Mut/MB. Similarly, we found an excellent correlation of TMB levels in all patient-PDX comparisons (MSI PDXs: 31.51 ± 16.95 Mut/Mb; HCN PDXs: 5.10 ± 1.92 Mut/Mb). As expected, genomic analysis of patient-PDX specimens showed that all specimens classified as MSI had an unstable microsatellite status, as opposed to HCN patients (Table 1, TMB and MSI status).

### 2.2. Molecular Analysis of PDXs

Next, we focused on uncovering the molecular landscape of PDX models by analyzing the WES data. As indicated in material and method section (Appendix A), WES data obtained from different PDXs derived from the same patient were integrated to generate a list of molecular alterations representative of an individual patient (Appendix A). First, we evaluated whether metastasis could be considered an integral part of the tumor or had they evolved and differed significantly from the primary tumor. Therefore, we compared metastatic tissue gene variants with primary tumor variants in the PDX models developed from metastatic areas (PDX548, 741, 782). We found that almost 100% of the SNVs present in primary tumor regions in the PDX, were also present in metastatic tissue (Appendix A), in addition to a set of metastasis-specific SNVs. Only one gene out of 381 SNVs was not included in the metastatic tissue from PDX741. Thus, this finding enabled us to integrate PDX models derived from metastatic tissue into our analysis to have a better coverage of the molecular alterations present in each patient. Additionally, this observation allowed the inclusion of patient #589, who was completely represented by metastatic PDX models.

To fully characterize the molecular landscape of PDX models, we first examined the number and type of SNVs of both groups (Appendix A). We observed a significant difference (*p* = 0.0023) in the number of genes carrying SNVs, showing a substantial increment in the MSI group compared with the HCN group: average <2000 SNVs vs. ~500 SNVs, respectively (Figure 2a). Next, we compared the gene lists and observed that 1611 genes overlapped between groups, representing only 16.7% of MSI genes but 67.4% of HCN genes (Figure 2b). However, when we analyzed the types of SNVs that occurred in each group, we found that the most frequent SNVs in the MSI groups were missense mutations, followed by frameshift mutations, representing almost 90% of all SNVs in this group (Figure 2c), while HCN PDXs contained mainly missense mutations (*p* < 0.01) (Figure 2d). Finally, we performed a hierarchical clustering heatmap considering the frequency of SNV events for each gene and confirmed that the mutational profile of the models differed significantly and can distinguish between MSI and HCN PDX groups (Figure 2e).

Somatic copy number variations or CNVs are another type of common occurring event in tumor cells. We analyzed both types of CNVs in our PDX models; GAIN events, representing the amplification of a genomic region, and LOSS events, constituting the deletion of a genomic region. For the CNV GAIN analysis, we considered only those amplified genomic regions with more than three copies, while all kinds of CNV LOSS events were taken into consideration. We observed that the MSI group presented significantly fewer CNV GAIN (*p* < 0.05) and LOSS events compared with HCN (Figure 3a). Despite some PDX models from the MSI group presenting CNV GAIN (PDX521, −526) and LOSS (PDX505, −516, −526) events comparable to the HCN group, when we analyzed the number of genes contained in the amplified or deleted genomic regions, we observed that the amplified regions contained relatively few genes (Appendix A). PDX526 is the only MSI model that behaved comparably to HCN as regards the frequency of CNV events and the number of genes contained in the amplified/deleted genomic regions. To delve further into this, we evaluated the frequency distribution of CNV events along chromosomes using Circos representation. We found a remarkable difference between the MSI and HCN groups; while HCN showed a homogenous distribution of CNV events along the different chromosomes, with a higher frequency of GAIN events, specifically focused on chromosomes 3, 7, and 8, and LOSS events on chromosome 5, 9, 13, 15, 18, and X; the MSI group solely showed a mild frequency of GAIN events in chromosomes 1 and 2, and LOSS events in chromosome X (Figure 3b). In addition, when we analyzed the potential tumorigenic role of CNV genes, we found that the number of amplified genes were comparable between groups, however, HCN PDX models exhibited a higher number of LOSS tumor driver genes (Appendix A). Specifically, MSI PDX models exhibited decreased total number of tumor driver genes compared with the HCN group (142 vs. 195 genes, respectively) with an overlapping of 79% of altered tumor driver genes from the MSI (112/142) in HCN genelist (Appendix A).

Altogether, our results showed that PDX models from EC patients exhibited specific molecular features associated with the molecular classification subtype they belong to.

### 2.3. PDX Validation with Patients

Next, we aimed to validate PDXs as preclinical models that reliably represent their corresponding EC patients by comparing WES data from PDX tumors with patient samples, either primary tumor or UA.

UA has been previously described as an EC tissue surrogate at transcriptomic and genomic levels [13,14,15]. To validate this, we compared genes carrying SNVs in primary tumor (PT) tissue and UA of two different patients, MSI:524 and HCN:596, representing one endometrioid-MSI and one serous-HCN patient (Figure 4a, blue and orange bars, respectively). As expected, the percentage of common genes carrying SNVs was higher than 70% between primary tumor tissue and the UA sample, confirming a high level of similarity between both samples. The number of genes carrying SNVs was higher in UAs from MSI patients compared with HCN patients in agreement with our previous observations in PDXs (Appendix A).

Then, we performed a comparative analysis studying the concordance between both primary tumor and UA samples, and the corresponding PDX tumor regarding conservation of driver mutation genes. We found that most of the driver genes were consistent between patient samples and PDX tumor (Figure 4b; Appendix A). For instance, in patient 524, 36 out of 45 driver mutations (80%) from primary tissue were shared with UA and 40 out of 45 with the PDX tumor (89%). Interestingly, those 36 driver mutations shared between primary tumor tissue and UA are the same ones shared between UA and PDX, supporting the hypothesis that UA is a valid surrogate of patient tissue (Figure 4b, left panel). Similarly, when analyzing patient 596, we found that 7 out of 13 driver mutation genes from primary tissue were also found in UA (54%), but representing 70% of UA driver mutations. Furthermore, of the six driver mutations shared between primary tissue and PDX, four (67%) were also found in UA (Figure 4b, right panel).

Once UA was confirmed as a high-fidelity sample to represent the molecular landscape of the patient, we compared SNV and CNV alterations in PDX models to their patient counterpart UA samples. PDXs derived from MSI tumors had a high degree of similarity to their patient counterparts for SNVs (mean value of 71%) and a moderate correlation with CNV GAIN and LOSS events (average 51% and 26%, respectively) (Figure 4c–e, left panel). On the contrary, PDXs derived from HCN tumors highly correlated with the UA-patient counterpart regarding CNV GAIN and LOSS (86% and 75%, respectively), but had a moderate similarity to SNV alterations, with a mean overlap of 46% (Figure 4c–e, right panel).

Taken together, our results show that PDX models reproduce the major alterations observed in patients, considering that SNV alterations are predominantly related to MSI patients and CNVs are more related to HCN patients.

### 2.4. PDX Validation with TCGA

Our data already supported the potential of PDXs as representative models of EC patients. However, to validate whether our PDX models recapitulate the molecular landscape of their specific molecular subtype, we compared our data to an extensive independent cohort of EC patients. We used the Uterine corpus endometrial carcinoma study from the TCGA PanCancer Atlas database (Study ID: ucec_tcga_pan_can_atlas_2018) (Appendix A).

Firstly, we compared SNVs from individualized PDX models to the TCGA endometrioid/MSI or serous/HCN datasets, respectively. We observed that PDX models from the MSI group presented a high similarity for genes with SNVs compared to TCGA data, showing an overall similarity percentage over 80% (Figure 5a, left panel). However, as expected, HCN PDX models had a lower percentage of commonly mutated genes (SNVs), showing an overall similarity of around 40% to TCGA patients (Figure 5a, right panel).

To discard any bias in our comparison, we decided to examine the frequency and the number of genes carrying SNVs of all TCGA´s molecular subtypes EC patients. We found that MSI and POLE subtypes encompassed an extremely high number of mutated genes showing a 95–98% overlap of genes between them, while the number of genes carrying SNVs in HCN and LCN subtypes was relatively smaller (Appendix A). Thus, to have a representative gene set for each group, we decided to select the top 1000 most frequent genes carrying SNVs for each EC subtype. We observed a marked decrease in the number of overlapping genes between groups; resulting in a 50% decreased of common genes between POLE and MSI subtypes (Appendix A). Similarly, we observed a specific subset of unique genes in each group, supporting our idea of having particular genes associated with each EC subtype (Appendix A, colored cells).

Then, we compared SNVs between each PDX group to the TCGA gene set for each EC subtype. We observed that 87% of the MSI subtype genes were present in our MSI PDX models, while this percentage decreased significantly when compared with the other subtypes (POLE 79%; HCN 71%; LCN 73%) (Appendix A, left panel). As expected, the level of SNV similarity was lower between HCN PDXs and all of the different EC subtypes. Moreover, our results showed that HCN PDXs were not specifically associated with SNVs from the HCN subtype since the range of similarity was similar for all EC subtypes (25–32%) (Appendix A, right panel).

Nevertheless, when we analyzed CNV genes, we observed the contrary situation. MSI PDX models showed the lowest percentage of similarity to the corresponding CNV gene set from TCGA, with an overall similarity of 7.27% in amplified genes (Figure 5b, left panel), and 3.86% (Figure 5c, left panel) in depleted genes. However, HCN PDXs exhibited higher levels of similarity between CNV GAIN and LOSS events compared with the gene set from the TCGA cohort, with an overall similarity of 31.9% (Figure 5b, right panel) and 14.4% respectively (*p* < 0.05 vs. MSI) (Figure 5c, right panel).

### 2.5. Beyond the Genomics: Molecular Pathways and Biological Process Associated with MSI and HCN PDX Models

Having thoroughly described our EC PDX cohort both at the histological and molecular level and associated its molecular landscape with that of the patient and the molecular subtype they belong to, we moved forward to identify the most relevant genes related to each group. We sought to potentially identify altered pathways that could be exploited for applying novel, alternative or targeted therapies in those patients who are refractory to treatment or have recurred.

To do so, we first analyzed the frequency of all of the genes carrying SNVs and CNVs in MSI and HCN PDXs and compared them to the TCGA gene set. We focused solely on SNV genes with frequency higher than 25%, amplified/deleted (CNV) genes present in at least three different PDX models, and those described as tumor drivers. As expected, the MSI PDX gene list contained more genes related to SNVs (85 genes) compared with HCN PDXs (6 genes). Nevertheless, when we analyzed CNV genes, we observed that MSI PDX models hosted only 16 tumor driver genes found in three out of seven PDX models, all of them located in chromosome 1 (q arm, 144,886,229:249,212,605—GRCh37/hg19). In contrast, we found 50 tumor driver genes, 36 amplified and 14 deleted, in at least three HCN PDX models, distributed along chromosomes 1, 3, 5, 7, 8, 12, 13, 19 and 20, and chr 11, 13, 15, 16, 18, 19 and X, respectively (Figure 6a).

We performed a functional enrichment analysis to uncover differences in pathways regarding the molecular status of the tumor. First, we analyzed our gene lists with gProfiler software to determine the GO molecular function (MF), and biological processes (BP) differentially associated with the distinct PDX groups (Figure 6b; Appendix A). We observed that most MF terms were differentially associated with one or other PDX model (Figure 6c). In particular, the HCN group was significantly associated with binding terms (GO:0005488), such as transcription factor binding (GO:0008134), DNA binding (GO:0003677), and protein kinase binding (GO:0019901); while MSI PDXs were significantly associated with GO terms related to catalytic activity (GO:0003824) and transporter activity (GO:0005215), as observed by GO network association (Figure 6c).

Regarding BP terms, we also found a differential pattern of GO terms associated with each group. We observed that the HCN group was enriched in terms associated with metabolic processes (GO:0008152) and cellular processes (GO:0009987), (Appendix A). The MSI group showed an over-representation of genes associated with cell adhesion (GO:0007155) and biological adhesion (GO:0022610). Moreover, genes from the MSI group were also associated with terms of cellular processes (GO:0009987), however, we observed a moderate overlap among cellular processes terms with genes from HCN (Appendix A).

Finally, we also analyzed biological pathways using REACTOME and we again observed a differential association of pathways when comparing both groups. On the one hand, we found that pathways associated with p53 were over-represented in the HCN group, as expected since p53 is one of the most frequently altered genes in this group, as well as pathways of transcription regulation and gene expression (Appendix A). On the other hand, the MSI group presented a heterogenous set of pathways with chromatin organization and histone modifications being the most representative, and PI3K/AKT activation was also represented in this group.

Altogether, these results illustrate the differential molecular behavior of MSI and HCN groups suggesting specific cellular functions and the activation of a particular set of pathways or genes, that could be exploited for the design of targeted therapies and, thus, for the development of personalized medicine.

## 3. Discussion

Generally, EC patients are diagnosed when the tumor is still confined to the uterus, which is associated with a favorable prognosis. However, 15–20% of the cases are diagnosed when the tumor has spread to other tissues, which is related to an increased incidence of distant metastases and recurrence [16]. In those patients at high risk of recurrence, treatment options are limited being the gold standard a chemotherapy treatment based on platin and taxanes. In this context, the development of highly representative EC models that molecularly and histologically recapitulate the patient´s disease might become an extraordinary preclinical tool for testing the efficacy of novel targeted therapies and approach personalized medicine. Here, we aimed to validate our PDX models of EC as a reliable preclinical tool that retains the molecular profile and histological features of EC patients. This study reports the ability of PDXs to resemble high-risk and recurrent EC patients with molecular profiles endometrioid/MSI or serous/HCN by using WES data and immunohistochemistry analysis.

First, we demonstrated the high concordance of the histological and molecular classification of EC patients with their PDX counterparts. We observed that our patient-PDX cohort matched perfectly according to histological classification, showing only one discrepancy in the molecular classification, due to misclassification of patient #521 as LCN by the ProMisE system. However, we observed that the patient exhibited amplification of microsatellite markers (BAT25, BAT26, NR-21, NR-24, and MONO-27) and, hence, was reconsidered as MSI. Interestingly, it is reported that 5–10% of MSI patients are misclassified due to IHC analysis, a PCR validation being necessary to accurately classified these patients [17,18]. Similarly, when we compared specific p53 and MMR protein expression between patients and their PDX models, we also observed some discrepancies that could be explained by the subclonal nature of EC tumors. It was reported by Singh et al. [19] that subclonal p53 immunostaining in endometrioid EC could be attributable to MMRd, and the subclonal nature of its expression likely reflects the acquisition of TP53 mutations as a later event during tumor progression.

Next, we demonstrated the robust differences between both EC groups, MSI and HCN, as regards their molecular profile. We found that MSI PDX models have a higher number of genes carrying SNVs, compared with HCN PDXs. We also found remarkable differences among the number of CNV events and the number of specific genes associated with these events. We observed that HCN PDXs had a higher frequency and a homogeneous distribution of CNV events along chromosomes compared with MSI PDXs, in which CNVs were localized to specific genomic regions. Interestingly, MSI PDXs were enriched in CNVs localized on chromosome 1 q-arm, containing the 1q32.1 region, which has been reported as a prognostic marker associated with worse relapse-free survival and higher risk prediction of recurrence in EC patients [20]. Indeed, our results confirmed previous observations reported by the TCGA consortium and others, demonstrating that both EC subtypes are genetically different, and must be considered separately for clinical management of patients despite the fact that they could both be classified as high-risk EC.

PDX models have emerged as an outstanding approach for translational research. Their ability to recapitulate key aspects of human malignancies by retaining histological and molecular markers from the patient turns them into a powerful tool for drug-testing assays and drug-response biomarker identification [4,21]. In this study, we compared all genes carrying SNV and CNV events between PDX tumors and their patient counterparts, using primary tissue samples or UA, to determine the degree of similarity between them. First, we validated UA as a valuable sample for typifying EC by comparing it to the patient’s primary tissue sample. We found that, independent of patient classification, UA shared 70% of SNVs with the primary tumor. Then, when contrasting WES data from PDX and UA, we confirmed that MSI PDXs showed a higher rate of similarity of SNVs to patients’ samples than HCN PDXs. However, this scenario was completely different for CNVs, in which HCN PDXs showed higher levels of similarity to UA samples among amplified and depleted genes (86% and 75% respectively) in comparison with MSI PDX, whose similarity levels dropped to 51% and 26%, respectively. These results again demonstrated the noticeable differences between the MSI and HCN groups regarding molecular alterations, each having a distinct genetic profile, in which retaining SNVs seems to be relevant for MSI PDXs to conserve its characteristic phenotype at the expense of CNVs, while HCN PDXs reliably reproduced the CNV profile but SNVs were not preserved alike.

At this point we questioned why the molecular similarity between PDXs and their patient counterparts was less than 90%. One possible explanation could be related to the patient sample collection process, PDX generation, and patient-derived tumor “evolution” in the mice. We must point out that, even if we collected different types of samples and different regions of primary or metastatic tissue, we would never have a 100% representation of the patient’s whole tumor. That is, unless we performed single-cell sequencing of the tumor from both the patient and the PDX, it would not be possible to capture and determine the entire genetic landscape of a patient´s disease. Hence, as previously demonstrated [15] and as we confirmed in this study, UA is the best and most feasible approach that we have at present to capture tumor molecular heterogeneity. In addition to UA, we developed PDX models for every EC patient from different areas of primary or metastatic tumor tissue, thus increasing the coverage of patient tumors. This contributed enormously to having a greater comprehension of the molecular landscape of a highly heterogenic tumor, but it was still not enough to guarantee full coverage of the patient’s molecular tumor profile. It is becoming clear that, despite that most of the patient’s biological characteristics are maintained in PDX models, tumors developed as PDX undergo mouse-specific evolution and can acquire new molecular alterations not present in the patient. Indeed, the patient-derived tumor undergoes selective pressure induced by the new murine environment, which exerts a regulatory function in tumor cells by their crosstalk with immune cells, endothelial cells, and stromal cells, and the tumor must adapt rapidly to this new environment, with replacing human blood vessels and stroma cells with murine cells being the most critical steps. Undoubtedly, the loss of the original tumor microenvironment and the acquisition of the new murine microenvironment is one of the reasons why the genotypic and phenotypic stability of the tumor is compromised in PDX models [22]. However, some studies suggest that, although there is indeed engraftment-associated selection, the majority of changes do not occur in oncogenic driver genes, therefore not affecting tumor biology [23]. Similarly, PDX tumor molecular evolution along time has also been monitored, finding that even when there is a disappearance of patient-specific somatic CNVs and acquisition of novel somatic CNVs over time in PDX, there is still an enrichment of CNVs in PDX models that correlates with patient primary tumor [24]. Our results reinforce these observations, since we found a conserved gene set present in all types of samples compared (Figure 4b) from both patient (PT tissue and UA) and PDX.

The Cancer Genome Atlas network has analyzed many human tumors to discover molecular aberrations at the genomic, transcriptomic, proteomic, and epigenomic levels. In this work, we selected the Uterine corpus endometrial carcinoma Pan-Cancer study to analyze to what extent our PDX models were represented in a more extensive cohort of EC patients. Our results demonstrated that PDXs classified as MSI showed a high degree of molecular similarity (87%) to the endometrioid MSI TCGA dataset when comparing genes carrying SNVs. However, the molecular similarity dramatically decreased when comparing amplified (7.3%) or deleted (3.9%) genes by CNV. In contrast, HCN PDXs showed a moderate percentage of similarity (40%) regarding SNV, and a modest similarity of 31.9% and 14.4% for CNV GAIN and LOSS respectively, when compared with TCGA. Despite the apparent similarity mismatch, mainly in the HCN group, our results were proven to be trustworthy considering that most patients included in the TCGA had distinct or unique alterations not shared with other samples in the same cohort [25]. Indeed, when we analyzed the TCGA dataset, we observed that the endometrioid MSI subtype had a defined pattern of genes with a higher frequency of SNVs than the serous HCN subtype, which seems to be represented by patients carrying randomly mutated genes, characterized by low-frequency SNVs. This, in part, could explain the differences observed when comparing SNVs from MSI and HCN with their respective TCGA dataset. Interestingly, we were able to demonstrate that MSI PDX models were highly represented by the TCGA cohort, while serous HCN seem to be more related to CNV.

Extensively characterized PDX models could represent a unique tool for the design of novel strategies to test drug efficacy [26] and could be used in preclinical trials to enhance therapy predictive value and ensure an increment of the success rate in drug development and safety translation to the clinic. However, PDX models also have limitations. Subcutaneous engraftment is the most common approach for PDX development used in personalized and targeted-therapy trials [11,27,28], however, more challenging approaches could be performed by using orthotopic engraftments, humanized murine models, and the design of preclinical trials including multiple, well-characterized PDX models (i.e., clinical trials). For instance, orthotopic tumor xenograft models provide a more appropriate physiological context to assess the disease, showing a greater capacity for growth and metastasis formation in comparison with subcutaneous models [29,30]. Next, by using humanized models, we can overcome the lack of immune microenvironment in PDX models, thus allowing us to preclinically evaluate the efficacy of immunotherapies. Finally, by performing mouse clinical trials, it would be possible to obtain a more realistic understanding of the population response to specific treatment and the identification of responding biomarkers [21].

In context to the aforementioned, the results obtained in our study, regarding the histological and molecular comparison of PDX models with their patient counterparts, must be taken as an excellent result demonstrating the ability of PDX models to significantly retain and recapitulate molecular features of the patient from which they were developed. This study was limited to MSI and HCN EC models, and thus, further research should also consider the study of LCN and POLE ECs to provide evidence on the availability of representative models of all EC molecular subtypes. Also, our study was limited to provide the correlation and potential of these models in terms of treatment response. The link with pharmacological studies of some of these EC PDX models was proven in previous publications of the group. Palbociclib was tested in EC PDX model 741, which was an MSI model bearing mutations in PTEN [11]; and we also tested ABTL0812, which is a novel small molecule inhibitor, in the HCN 548 and the MSI 521 EC PDX models [12]. The results derived from these preclinical studies have already contributed to move forward personalized treatment for EC patients. Specifically, thanks to the preclinical studies on ABTL0812, a phase 2 study (NCT03366480) was conducted to assess the efficacy of this treatment in recurrent or metastatic EC patients.

## 4. Materials and Methods

### 4.1. Patient Inclusion Criteria and Sample Collection

This was a retrospective study including patients with a final diagnosis of endometrioid EC with FIGO Stage IB or higher and histological grade 2 or 3; or with a non-endometrioid EC; all of them were women above 45 years of age. In this study, patients having at least a successful PDX model from two different areas of the primary tumor (PT), metastasis, or recurrence were selected.

From each patient we collected endometrial uterine aspirate (UA), whole blood sample, tissue samples from normal endometrium, and tumor tissue from different regions of the primary tumor, and from local or distant metastasis, if available.

UAs were collected by aspiration with a Cornier Pipelle in the operating room prior to surgery and processed as described [13]. Similarly, peripheral blood samples were also collected before surgery and a two-step centrifugation protocol was performed to separate plasma, buffy coat and pellet fractions as previously described [13]. All tissue samples were macroscopically collected by an experienced pathologist from the Pathology department of Vall d’Hebron Hospital and stored at −80 °C or used for the development of PDX models. Medical records, as well as clinicopathological data, were also available in a dissociated and pseudo-anonymized manner.

In this study only 13 patients (Table 1) were able to reach WES quality standards and subsequently were used for the comparison to its PDX counterparts.

All patients included in the study signed an informed consent form before any intervention accepting the transference of biological material or clinical data for research use. The study was conducted in accordance with the Declaration of Helsinki, following all of the requirements established by the Ethical Committee for Clinical Investigation (CEIC—Procedure approved №: PR(AMI) 276/2018) of Vall d’Hebron Hospital, regarding national and international guideline regulations on data protection and confidentiality.

### 4.2. PDX Generation

All PDX models were developed by subcutaneous implantation of tumor samples from EC patients into athymic nude mice (6-week-old female Swiss nu/nu). Briefly, small pieces of fresh primary or metastatic tissue were subcutaneously implanted in both flanks of anesthetized mice. Tumor development was monitored weekly until it reached a volume of 1000 mm^3^. Then, tumors were excised, fractionated and stored fresh-frozen, formalin-fixed and paraffin-embedded, biobanked in DMSO-Serum solution for cryopreservation and/or passed to another mouse for PDX amplification and tumor propagation [4].

PDX cohort was generated from different anatomical areas of the primary tumor (PT), a superficial (cell-layer close to the uterus cavity), a deep (invasive front, close to the myometrium) tumor area, and in some cases, from metastatic tissue (regional or distantly located). Each tumor area was implanted in one mouse (generation 0). This strategy enabled us to capture and cover a wide range of patient´s tumor alterations into a set of mice carrying a distinct tumor area. Appendix A summarizes patient’s tumor origin for PDX development. Thus, each PDX carried with a unique tumor area from EC patient resulting that each patient was represented by one or more PDX models. Despite this, here we integrated all of the molecular and histological data of the different PDX models into just one subject representing the patient for their analysis.

All procedures involving animals were previously approved by the Ethical Committee for Animal Research (CEEA, Approved №: CEA-OH/10533/1) at the Vall d’Hebron Institute of Research in accordance with national and international guidelines for animal welfare.

### 4.3. Whole-Exome Sequencing (WES) Data Generation

DNA from each EC patient’s peripheral blood, UA, tissue samples, and paired PDX model tumor samples was isolated for WES analysis (Table 2). DNA purification from peripheral blood samples was performed using the Puregene Blood Core Kit A (Qiagen, Minneapolis, MN, USA), and the AllPrep DNA/RNA Mini Kit (Qiagen, Hilden, Germany) was used for UA and tumor tissue samples. WES was performed using the Roche CSP Nimblegen exome capture kit and Illumina HiSeq 4000 sequencing platform at the National Centre for Genomic Analysis (CNAG, Barcelona, Spain), with a mean coverage of 150× (2 × 100 pb).

For the bioinformatics analysis, reads from WES were aligned to the GRCh37/hg19 human reference genome. Somatic SNVs were identified using MuTect2 from GATK (version 4.1.2.0) [31] and small insertion and deletions (indels) using Strelka2 (version 2.8.3) [32] and VarScan 2 [33]. The potential functional effect of each change was assessed using a combination of multiple predictors. Somatic CNVs, including loss of heterozygosity, were studied using TITAN [34]. The cancer cell fraction was calculated using ABSOLUTE [35]. The mutational signature of each sample was obtained by the analysis of the mutation classified in C > A, C > G, C > T, T > A, T > C or T > G and then subcategorized according to the nucleotides preceding (5′) and succeeding (3′) the mutated base.

For tumor mutational burden (TMB) analysis, non-synonymous somatic variants in coding regions were considered in those samples with at least 10 reads in both tumor and reference samples.

For genomic MSI analysis determination in WES data, MANTIS (Microsatellite Analysis for Normal Tumor Instability) software was run using WES recommended default parameters [36].

The total number of SNVs and CNVs for each PDX was obtained by analyzing all of the individualized PDX models for each patient and integrating all of the information retrieved for a specific patient, including all SNVs and CNVs that were identified in at least one of the tumor areas.

### 4.4. Molecular Classification of EC Patients and PDX Tumors

To molecularly classify EC patients and their corresponding PDXs, we followed The Proactive Molecular Risk Classifier for Endometrial Cancer (ProMisE) surrogate system [37,38], which interrogates MMR proteins and p53 expression by immunohistochemistry determination, and evaluates somatic mutation of POLE by DNA sequencing.

We performed immunohistochemistry analysis on formalin-fixed paraffin-embedded (FFPE) tissues from EC patients and their corresponding PDX models. Hematoxylin-eosin (H & E) and immunohistochemistry staining of MMR proteins (PMS2, MSH2, MLH1, MSH6) and p53 were performed and blindly analyzed for molecular classification. Briefly, immunohistochemistry staining for p53 and MMR proteins was performed in 5 µm FFPE tissue slides using a fully automated system, the Benchmark ULTRA slide Stainer (Ventana Medical Systems, Tucson, AZ, USA). The slides were deparaffinized (EZ prep TM (10×), Ventana Medical Systems), and antigen retrieval was performed by incubating with cell conditioning solution (pH: 8, Ventana Medical Systems), blocked with hydrogen peroxide solution (3%) and rinsed with reaction buffer (10×, Ventana Medical Systems, Tucson, AZ, USA). The slides were then incubated with the primary antibody p53 ((DO-7), mouse monoclonal #800–2912) at 37 °C for 44 min, MLH1 ((M1), mouse monoclonal #760–5091), MSH6 ((SP93), rabbit monoclonal #760–5092) and MSH2 ((G219–1129), mouse monoclonal #760–5093) at 37 °C for 40 min, PMS2 ((A16-4), mouse monoclonal #760–5094) at 37 °C for 92 min, followed by amplification using an UltraView Polymer Detection Kit (Ventana Medical Systems, Tucson, AZ, USA) with diaminobenzidine as the chromogen. Finally, slides were counterstained with hematoxylin and deparaffinized. Additionally, in those patients whose MMR protein profiles were undetermined or discordant between patient and PDX, MSI status was verified by testing the amplification of microsatellite markers BAT25, BAT26, NR-21, NR-24 and MONO-27 through the MSI Analysis System Kit (PROMEGA) following manufacturer’s instructions [17].

POLE mutations were determined by Polymerase Chain Reaction (PCR) amplification of the POLE gene (exons 9, 11, 13, 14), and subsequent genotyping using KASP technology V4.0 2× to identify five of its most common hotspot mutation sites (P286R, S297F, V411L, A456P, S459F) (GC Genomics/KBioscience, reference KBS-1016-021) [39]. DNA extraction was performed on five 5 µm sections from FFPE specimens using the QIAamp^®^ DNA FFPE Tissue Kit (QIAGEN, #56,404) following manufacturer’s instructions. POLE mutations were also confirmed by WES analysis.

Interpretation of molecular classification staining was carried out by an expert pathologist from the Pathology Department of the Vall d’Hebron Hospital. MMR proteins were identified as abnormally (abn) expressed when no nuclear staining was detected in one or more of the MMR proteins. Aberrant p53 expression was recognized as overexpression (tumor cell nuclei stained at an intensity higher than 75%) or no expression of the protein in the tumor cell nuclei [19,40].

### 4.5. WES Data Analysis, Interpretation, and Visualization

We used different software for WES data analysis and graphical representation. Venn diagrams were used to group datasets from SNVs and CNVs using the webtool from Ghent University (http://bioinformatics.psb.ugent.be/webtools/Venn/ (accessed on 10 June 2021)). The visualization format of the Venn diagrams was developed by eulerr web tool (http://eulerr.co/ (accessed on 10 June 2021)) [41]. A hierarchical clustering heatmap was performed using R (http://www.R-project.org/ (accessed on 18 June 2021)) [42] for the analysis of the frequency of genes carrying SNVs in PDX tumor samples.

Cancer genome interpreter (CGI, https://www.cancergenomeinterpreter.org/ (accessed on 3 March 2021)) [43] was used to analyze SNV and CNV amplified or deleted genes classified according to their tumorigenic role (tumor driver or passenger mutations).

CNVs of all PDX samples were analyzed and visualized using Circos [44]. For CNV GAIN, amplified regions with more than three copies were considered, while all CNV LOSS regions were analyzed.

For pathway enrichment analysis, we used gProfiler (https://biit.cs.ut.ee/gprofiler/gost (accessed on 20 June 2021)) [45] and Cytoscape software [46].

### 4.6. TCGA Dataset Analysis

TCGA datasets were accessed via cBioPortal (https://www.cbioportal.org/ (accessed on 21 February 2021)) [47]; The uterine corpus endometrial carcinoma dataset (TCGA, PanCancer Atlas, Study ID: ucec_tcga_pan_can_atlas_2018) was chosen from all of the available EC datasets due to it presenting the most detailed and complete patient information. Genes carrying SNVs and CNVs from Endometrioid-MSI, LCN, POLE, and Serous-HCN groups of patients were selected for comparison with our data.

### 4.7. Statistical Analysis

Statistical analysis was performed with GraphPad PRISM Software 6.0, by using non-parametric tests such as U-Mann Whitney (comparisons between two groups) or Kruskall-Wallis for multiple comparison test (comparisons between more than two groups). Statistical significance: ns = not significant; *, *p* < 0.05; **, *p* < 0.01; ***, *p* < 0.001.

## 5. Conclusions

Among the challenges facing the application of PDXs in the preclinical evaluation of anticancer drugs, the development of well-grounded models that reliably represent the patient is crucial. We have demonstrated that our PDX models efficiently recapitulate, molecularly and histologically, the characteristics of the EC patients from which they originated; and we also showed that PDXs exhibited a good correlation with the molecular data from the EC patients recruited and analyzed by the TCGA network. Despite the well-known limitations that PDXs present, the results obtained in this study demonstrated that EC PDXs can be considered robust and reliable preclinical models.

## Figures and Tables

**Figure 1 ijms-23-06266-f001:**
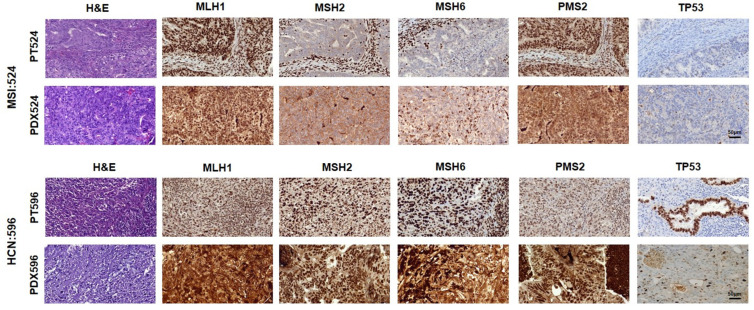
Patient-PDX histological characterization. Tumor tissue staining of a representative endometrioid/MSI EC patient and their PDX model (MSI:524, (**upper panel**)) and a serous/HCN EC patient and their PDX model (HCN:596, (**lower panel**)). Images show the hematoxylin and eosin (H & E) staining, as well as IHC of MMR proteins MLH1, MSH2, MSH6, PMS2, and TP53. Magnification 40×.

**Figure 2 ijms-23-06266-f002:**
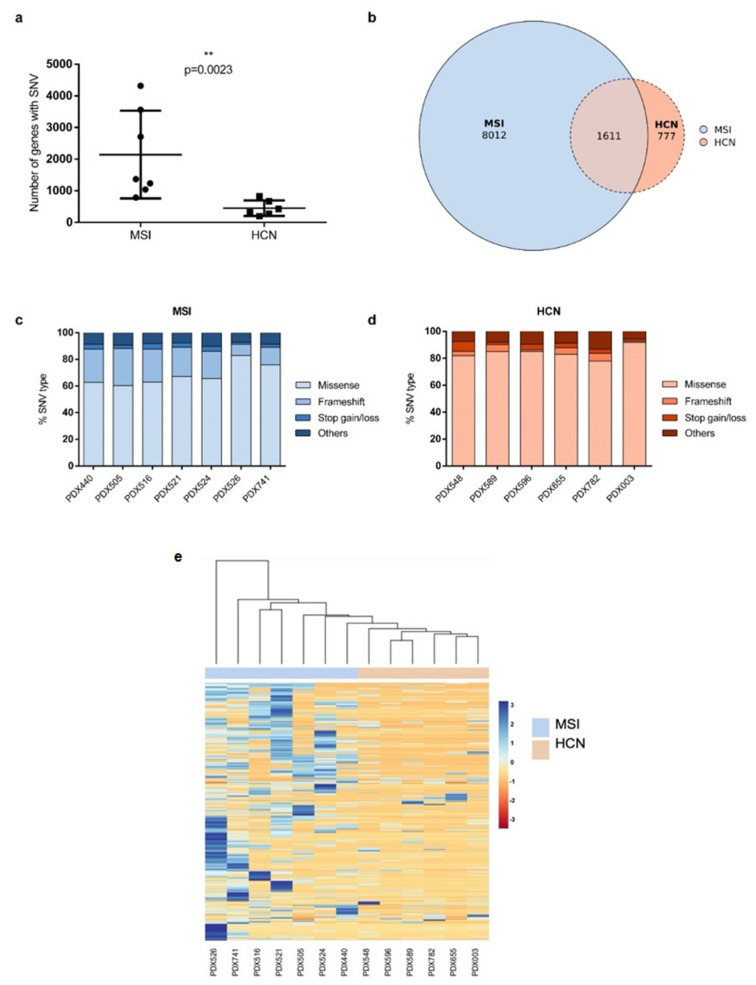
Molecular characterization of PDX models based on single-nucleotide variants. (**a**) Comparative analysis of the total number of genes carrying SNVs in each PDX model, grouped as MSI and HCN. (**b**) Venn Diagram showing the number of genes with SNVs for each PDX group, MSI (blue) vs. HCN (orange) and the overlap between them. (**c**,**d**) Percentage of the most frequent types of SNV found in MSI and HCN PDX models. Others classification included all minority SNVs such as spliced donor/acceptor variant, start lost or any other alteration that did not fit with indels, missense or stop gain/loss mutations. (**e**) Hierarchical heatmap clustering analysis based on the frequency of genes carrying SNVs for both MSI and HCN PDX models.

**Figure 3 ijms-23-06266-f003:**
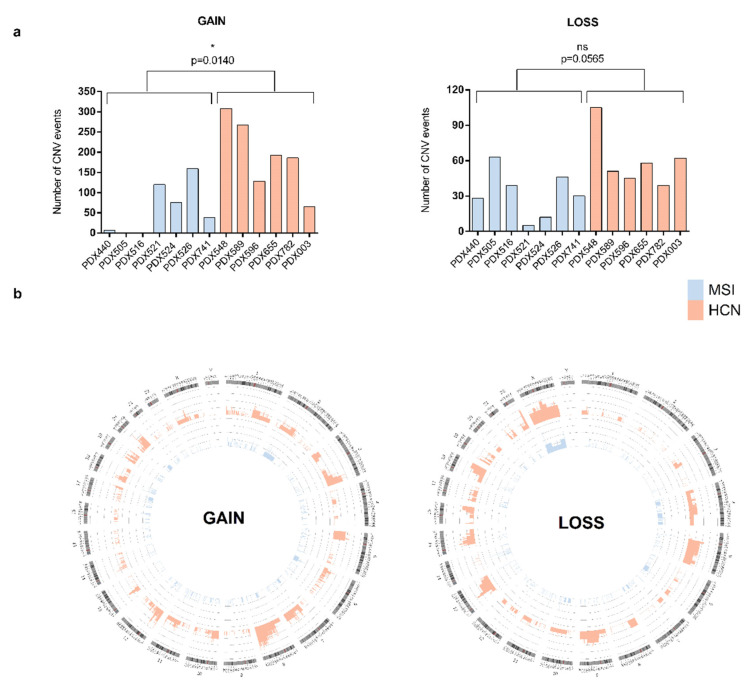
Molecular characterization of PDX models based on copy-number variants. (**a**) Comparative analysis of the total number of CNV events in each PDX model, grouped as MSI and HCN. Left panel shows the number of genes included in CNV GAIN events, while right panel reveals the number of genes included in CNV LOSS events. (**b**) Circos representation of the frequency and genomic distribution of CNV GAIN (**left panel**) and LOSS (**right panel**) events for each PDX group, MSI (blue, inner circle) vs. HCN (orange, outer circle).

**Figure 4 ijms-23-06266-f004:**
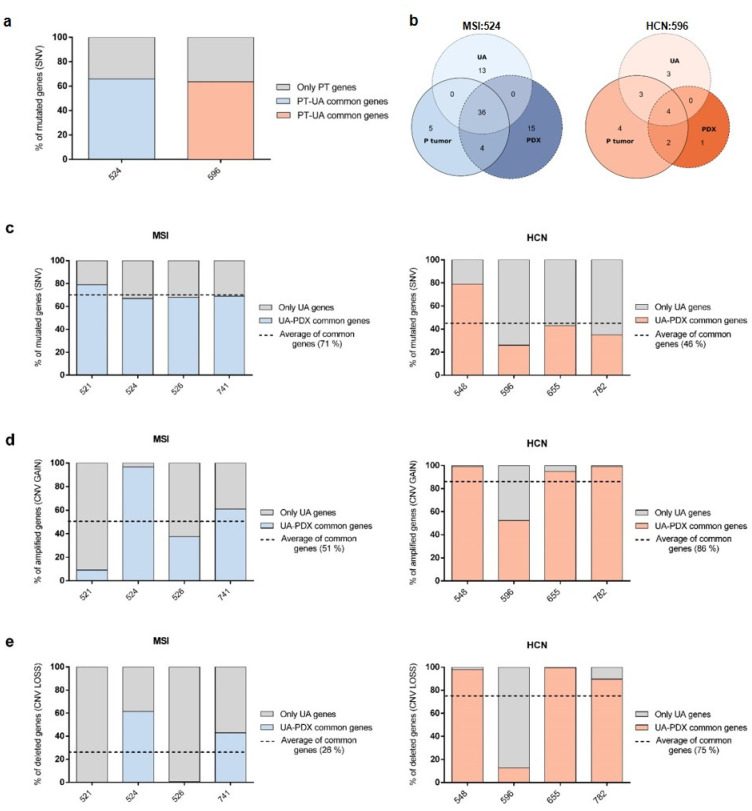
Molecular comparative analysis between PDXs and their EC patient counterparts. (**a**) Percentage of common genes carrying SNVs between primary tumor (PT) and UA samples from two representative EC patients, endometrioid/MSI (patient 524, blue-grey), and serous/HCN (patient 596, orange-grey). Colored bars represent the percentage of common genes between samples (PT vs. UA, blue: MSI:524; orange: HCN:596), while grey bars show the unique SNV genes present solely in the PT sample. (**b**) Venn diagram representation of the tumor driver genes found altered in MSI:524 (**left panel**) and HCN:596 (**right panel**) patient samples (PT and UA) and PDX tumor. The number of specific or shared genes is shown for each type of comparison. (**c**) Bar-plots showing the percentage of common mutated genes carrying SNVs for each PDX model compared to UA samples. (**d**,**e**) Comparison of the percentage of genes associated with CNV GAIN (**d**) or LOSS (**e**) events in PDX tumor vs. UA samples. Colored bars (MSI, blue; HCN, orange) represent the percentage of common genes between the PDX tumor sample and the corresponding patient’s UA, while grey bars show the unique SNV/CNV genes present only in the UA sample. Dotted lines represent the mean value of the common genes identified for each comparison.

**Figure 5 ijms-23-06266-f005:**
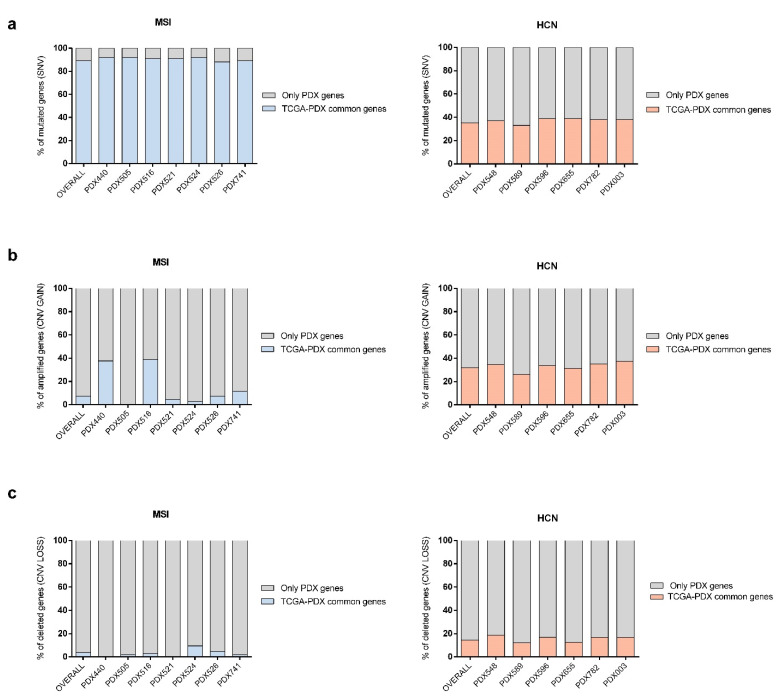
Comparative analysis of PDX tumor molecular variants and EC patient from the TCGA database. (**a**) Bar-plots showing the percentage of common mutated genes carrying SNVs for each PDX model compared to the endometrioid/MSI ((**left panel**), blue) or serous/HCN ((**right panel**), orange) patients from the PanCancer Atlas study (TCGA consortium). Grey bars show the percentage of unique SNV genes present solely in the PDX samples. (**b**,**c**) Comparison of the percentage of genes associated with CNV GAIN (**b**) or LOSS (**c**) events in PDX tumor. vs. patient’s from the above mentioned TCGA study. Colored bars (MSI, blue; HCN, orange) represent the percentage of common genes between the PDX tumor sample and patient from TCGA, while grey bars show the unique SNV/CNV genes present only in the PDX tumor sample.

**Figure 6 ijms-23-06266-f006:**
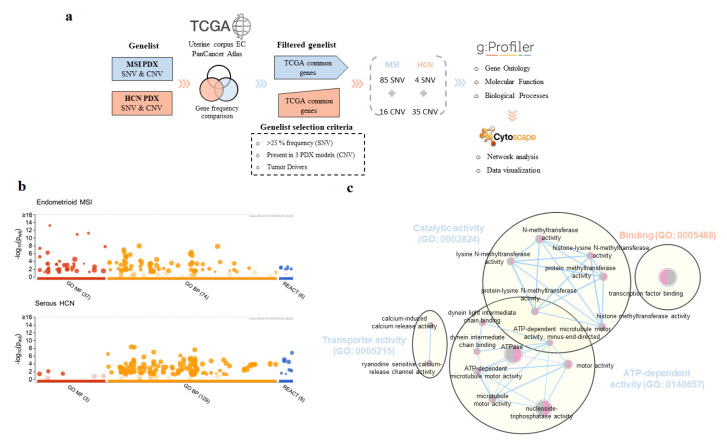
Molecular network analysis for the identification of relevant pathways associated with MSI or HCN profiles. (**a**) Schematic representation of the workflow followed for the identification of the most relevant genes carrying SNVs or involved in CNV events. A list of 101 genes (MSI), or 39 genes (HCN) were determined for each PDX group, according to the gene selection criteria described. (**b**) Molecular function (MF), biological process (BP) and reactome pathways (REACT) functional enrichment analysis of the most relevant genes of MSI and HCN PDX models. The plots show GO terms associated with a specific gene list for MSI (**upper panel**) or HCN (**lower panel**) PDX models. (**c**) Gene ontology (MF) network of the most relevant genes associated with MSI (blue terms) or HCN (orange terms). GO terms network analysis was performed by Cytoscape and terms were summarized using the AutoAnnotate application selecting a Q-value < 0.01 and a combined coefficient of 0.375.

**Table 1 ijms-23-06266-t001:** Clinicopathological and molecular features of EC patients and their PDX model counterparts.

Patient				Histological Classification					Molecular Classification								
Sample Code	Age	Risk	Recurrence	Histology	Grade	FIGO Stage	Myometral Invasion	LVSI	p53	MSH6	MSH2	MLH1	PMS2	POLE	TCGA	TMB (Mut/MB)	MSI Status
PT440	75	High	No	Endometrioid	2	II	<50%	Yes	WT	WT	WT	Abn	Abn	WT	MSI	-	-
PDX440				Endometrioid	2				WT	WT	WT	Abn	Abn	WT		High (14.04)	Unstable
PT505	52	Low	Yes	Endometrioid	1	Ia	<50%	No	WT	WT	WT	Abn	Abn	WT	MSI	High (15.47)	Unstable
PDX505				Endometrioid	1				WT	WT	WT	Abn	Abn	WT		High (14.14)	
PT516	83	High	Yes	Endometrioid	3	Ib	>50%	Yes	WT	WT	WT	Abn	Abn	WT	MSI	High (86.73)	Unstable
PDX516				Endometrioid	3				WT	WT	WT	Abn	Abn	WT		High (21.27)	
PT521	57	High	No	Endometrioid	2	IIIa	<50%	No	WT	WT	WT	WT	WT	WT	LCN *	High (45.56)	Unstable
PDX521				Endometrioid	2				WT	WT	WT	Abn	Abn	WT	MSI	High (43.52)	
PT524	38	High	No	Endometrioid	3	II	<50%	Yes	WT	Abn	Abn	WT	WT	WT	MSI	High (31.32)	Unstable
PDX524				Endometrioid	3				WT	Abn	Abn	WT	WT	WT		High (24.99)	
PT526	68	High	No	Endometrioid	3	Ib	>50%	No	WT	WT	WT	WT	Abn	WT	MSI	High (20.65)	Unstable
PDX526				Endometrioid	3				Abn	WT	WT	WT	Abn	WT		High (55.82)	
PT741	78	High	No	Endometrioid	2	IIIc2	>50%	Yes	WT	WT	WT	Abn	Abn	WT	MSI	High (28.47)	Unstable
PDX741				Endometrioid	2				-	WT	WT	Abn	Abn	WT		High (46.80)	
PT548	73	High	No	Serous	3	IIIc2	>50%	Yes	Abn	WT	WT	WT	WT	WT	HCN	Low (11.98)	Stable
PDX548				Serous	3				Abn	WT	WT	WT	Abn	WT		Low (7.90)	
PT589	57	Intermediate	Yes	Serous	3	Ia	<50%	No	Abn	WT	WT	WT	WT	WT	HCN	-	-
PDX589				Serous	3				Abn	WT	WT	WT	WT	WT		Low (4.89)	Stable
PT596	74	Intermediate	Yes	Serous	3	Ia	<50%	No	Abn	WT	WT	WT	WT	WT	HCN	Low (1.69)	Stable
PDX596				Serous	3				Abn	WT	WT	WT	WT	WT		Low (4.70)	
PT655	66	High	No	Serous	3	IIIc2	>50%	Yes	Abn	WT	WT	WT	WT	WT	HCN	Low (4.39)	Stable
PDX655				Serous	3				Abn	WT	WT	WT	Abn	WT		Low (3.27)	
PT782	81	High	No	Serous	3	IIIc2	>50%	Yes	Abn	WT	WT	WT	WT	WT	HCN	Low (3.16)	Stable
PDX782				Serous	3				Abn	WT	WT	WT	Abn	WT		Low (3.04)	
PT003	89	High	-	Serous	3	IV	-	-	Abn	WT	WT	WT	WT	WT	HCN	-	-
PDX003				Serous	3				Abn	WT	WT	Abn	WT	WT		Low (6.79)	Stable

* Patient classified as LCN following ProMisE system (IHC characterisation) and re-classified by PCR analysis of microsatellite markers. PT: patient, LVSI: lymphovascular space invasion, POLE: polymerase epsilon, TCGA: The Cancer Genomic Atlas, TMB: tumor mutational burden, MSI: microsatellite instability, HCN: high copy number, WT: wild type, Abn: aberrant/abnormal.

**Table 2 ijms-23-06266-t002:** Type of samples collected and processed from patient and PDX for DNA extraction and WES analysis.

Model		440	505	516	521	524	526	741	548	589	596	655	782	003
Patient	Peripheral blood sample	Yes		Yes	Yes	Yes	Yes		Yes	Yes	Yes	Yes	Yes	Yes
	Uterine Aspirate				Yes	Yes	Yes	Yes	Yes		Yes	Yes	Yes	
	Non-tumoral tissue		Yes					Yes						
	Primary tumor tissue		Yes			Yes					Yes			
PDX	Tumor tissue	Yes	Yes	Yes	Yes	Yes	Yes	Yes	Yes	Yes	Yes	Yes	Yes	Yes

## Data Availability

Uterine corpus endometrial carcinoma dataset (TCGA, PanCancer Atlas, Study ID: ucec_tcga_pan_can_atlas_2018) was accessed via cBioPortal (https://www.cbioportal.org/ (accessed on 21 February 2021)).

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
