# Peer review of "Genomic Validation of Endometrial Cancer Patient-Derived Xenograft Models as a Preclinical Tool"

_ijms, 2022, doi:10.3390/ijms23116266_

Round 1

Reviewer 1 Report

The manuscript reports the molecular characterization of PDXs dervied from EC. The study follows other publications by th group on the matter, giving a comprehensive analysis of some features of these tumors.

The reults presented clearly show that PDXs generated well recapitulate the human tumor from which they originate. The molecular and histological concordance are very high and further support the potency of these models for preclinical studies which potentially could speed up the clinical traslation of the results. This is in line with several other manuscripts presenting similar results for other human tumors and corresponding PDX

I do not have any specific comments on the results reported a part from the lack of any correlation and potnetial of these model in terms of response to treatment. The group already presented activity in vivo in previous manuscritpm and palbociclib was studied in one model that is present also in this list.

Having a characterization from a molecular and biological point of view, perhaps some data on the response to treatment would have strngthened the results. The last part of the discussion is centered on this and i wonder why the authors (since palbociclib manuscript was publised 6 years ago) did not attempt to investigate the activity of drugs based on the molecular features they described.

Author Response

We thank the reviewer’s comment since he/she understood perfectly the data presented here which shows the high potency of these models for preclinical studies as they recapitulate the patient’s tumor as well as overall molecular landscape of EC described by large consortiums, i.e. the TCGA.

The reviewer pointed that the paper could have been further improved by including the  potential of these models in terms of treatment response. As he/she indicated, we already published some of our results in this line: Palbociclib was tested in one of our EC PDX models and was described in Dosil MA. et al. J Pathol 2017;  we also tested ABTL0812, which is a novel small molecule inhibitor, in a serous and an endometrioid EC PDX models in Felip I, Moiola CP. et al. Gynecol Oncol 2019; and more recently, we have published a study in which we assessed the genetic landscape of an EC patient to guide the selection of a potential personalized therapy, which was subsequently validated in five different PDX models generated from different areas of primary and metastatic tissue of the patient (Mota A. et al. Oncogene 2022). Those articles are now referenced in the present manuscript in the discussion section in order to accommodate the reviewer comment to connect EC tumor molecular status to the potential use of those to approach personalized medicine.

Reviewer 2 Report

The Authors describe the histological architecture and the molecular profile of endometrial cancer tissues obtained upon xenograft of clinical specimens in immunocompromised mice. The aim of the work is of interest as the number of PDX models for the study of endometrial cancers are still scanty.

The Introduction provide sufficient background to the reader. The Methods section is accurately written.

Results provide evidence that PDX models recapitulate the clinical situation, although pharmacological studies and data on the metastatic potential are still needed. Representation of results is not always clear:

- The characteristics of the case series are not exhaustively listed: Table 1 should indicate the site of origin of the clinical samples, which patients donated both primary and secondary tumor tissue, the number of animals used per each patient and per each kind of tissue

- Color legend in Fig 4a should be explained or corrected;

- genes with SNV, deletions, amplifications etc. should be listed in Excel files;

- gene sets obtained from TCGA should be indicated using identifiers and references, and briefly described;

- the list of driver genes should be indicated in Excel files;

- Figure 6 does not appear in the manuscript

- Box plot in Fig S3 is not appropriate to represent data as N<5 observations, and statistical test is not feasible accordingly.

In the Discussion section the Authors should cite and comment the paper by Bonazzi et al 2022 (Genome Med. 2022 Jan 10;14(1):3. doi: 10.1186/s13073-021-00990-z.)

Author Response

Answer: I appreciate reviewer comments to highlight the efforts that we have done to generate knowledge on the field of EC PDX models, which we agree that are scanty. We also agree with his/her comment on the need to include pharmacological studies and data on the metastatic potential.

Regarding pharmacological studies, and as stated in our previous reply to reviewer 1, we have included the pharmacological studies that have been done with our EC PDX models and that are published. This is now in the discussion section to support the use of those models for preclinical studies. Indeed, the results derived from preclinical studies using EC PDX models have already contributed to move forward personalized treatment for EC patients. Specifically, thanks to the preclinical studies on ABTL0812 that we published in  Felip I, Moiola CP. et al. Gynecol Oncol 2019, a phase 2 study (NCT03366480) was conducted to assess the efficacy of this treatment in recurrent or metastatic EC patients.

Regarding the data on the metastatic potential of PDX models, it is clear that the subcutaneous engraftment precludes this understanding and to overcome this drawback, orthothopic implantation of patient-derived tumors is a clear opportunity to fully understand the impact of treatments to reduce/inhibit tumor growth and invasive capacities. This drawback is included in the discussion section. Also, the understanding on the metastasis process can also be approached by the development of PDX models from metastatic tissue. At this respect, our study included PDX models from metastatic tissue, which interestingly contained almost 100% of primary tumor SNV and a unique set of gene variants. This result enabled us to integrate PDX models derived from metastatic tissue into our analysis to have a better coverage of the molecular alterations present in each patient. Alternatively, those acquired novel SNV could be studied in an independent manner to understand how those specific alterations are beneficial for the tumor to develop metastasis, including migration and invasion needed for tissue colonization. This is a very interesting topic that will be addressed in future research.

Representation of results is not always clear:

- The characteristics of the case series are not exhaustively listed: Table 1 should indicate the site of origin of the clinical samples, which patients donated both primary and secondary tumor tissue, the number of animals used per each patient and per each kind of tissue

Answer: We thanks reviewer for this comment. We revised the manuscript version and table 1, and we found some mistakes when referencing tables that could have also contributed to misperception of data exposed in each table. Table 1 compiles patient´s clinicopathological data (Age, tumor histology, FIGO stage, Grade, LVSI, myometrial invasion, recurrence), and the corresponding data from its PDX. Next, we compared molecular data of patient-PDX regarding EC molecular classification markers and genomic data (TMB & genome instability).

We have included an additional table showing the “characteristics of patient tumor tissue origin for PDX development”, in Supplementary Information (Table S2). This table compiles information from EC tumor areas collection being superficial or deep tumor area from EC patient, and in case of available additional tissue from the metastasis site. All collected tissues were implanted individually in different mice to develop PDX models.

Finally, Table 2 (Material and Methods Section 4.3) shows the “type of samples collected and processed from patient and PDX for DNA extraction and WES analysis”. The table summarize the type of samples recruited, collected and processed for WES analysis and used for comparisons.

Thus, altogether, Table 1, S2 and 2 summarize clinicopathological features of patient/PDX, molecular characteristics, types of samples collected and their used in the present project.

Now, Table S2 includes information to answer to reviewer request of “which patients donated both primary and secondary tumor tissue and the number of animals used per each patient”. We have also revised and corrected tables references and modified the paragraph in section 4.2 (PDX generation) to support this answer. We now think the information contained in the manuscript and tables are clearly exposed.

- Color legend in Fig 4a should be explained or corrected

Answer: Bars in figure 4a represent the percentage of altered genes in primary tumor (PT) samples from patients MSI:524 or HCN:596 and the overlap with those altered genes in the uterine aspirate (UA) counterpart. We show the percentage of genes carrying SNV only in the PT sample in grey colour, and we represent the percentage of genes carrying SNV in both samples PT and UA for MSI:524 or HCN:596 patients in blue and orange, respectively. Figure 4a legend caption was revised and modified accordingly to reviewer comments. Additionally, we introduced some modifications in the main text to clarify the interpretation of the figure (section 2.3, lines 6-8). Similarly, to reduced redundancy in the nomenclature used for patient ID and primary tumor samples, we have replaced in the whole text every PT abbreviation next to patient ID to #, and we left PT as indicative for primary tumor.

Additionally, we have replaced headings of Figure 4b to fit with the correct nomenclature used in the manuscript (MSI:524 and HCN:596, instead PT/PDX524, and PT/ PDX596, respectively).

- genes with SNV, deletions, amplifications etc. should be listed in Excel files

- the list of driver genes should be indicated in Excel files

Answer: Following reviewer suggestion, we have built tables containing a complete genelist of SNV, CNV, and identifing which alterations are considered as tumor drivers (Annex A Tables 1-3). We have referenced this table in the manuscript accordingly.

Additionally, we have revised and chage data from Figure S2b and c, due to misinterpretation and underestimateion of the amount of tumor drivers gene included in CNVs event in Endometrioid/MSI PDX models.

Figure S2c shows the number of genes amplified (AMP, left bars) or deleted (LOSS, right bars) due to CNV in MSI (blue) or HCN (orange) PDX models. We decide to split bars and show AMP and LOSS genes independently, to exhibit the real amount of genes included in CNV events in each group. Similarly, we have modified Venn diagrams accordingly (Figure S2c). Venn diagram shows solely the number of genes in each group independently of being amplified or deleted.

- gene sets obtained from TCGA should be indicated using identifiers and references, and briefly described

Answer: We agree with the reviewer and apologized for missing this information. We have included now the study ID used for the comparison of genes from our models to the dataset obtained from c-bioportal repository. The study ID (ucec_tcga_pan_can_atlas_2018) is now indicated in the Material & Method section.  

- Figure 6 does not appear in the manuscript

Answer: Sorry if Figure 6 was not uploaded properly. We have uploaded and checked that the figure is now in the compiled PDF.

- Box plot in Fig S3 is not appropriate to represent data as N<5 observations, and statistical test is not feasible accordingly

Answer: We agree with the reviewer, and we modified accordingly this figure. We just left the dispersion dots to show the qualitative differences among groups, without any statistical support. In addition, the result section was modified accordingly.

In the Discussion section the Authors should cite and comment the paper by Bonazzi et al 2022 (Genome Med. 2022 Jan 10;14(1):3. doi: 10.1186/s13073-021-00990-z.)

Answer: We agree with the reviewer that Bonazzi et al article is important and support our data. We have cited the paper and contextualized it within the significant applications of molecularly-characterized PDX.

Reviewer 3 Report

In this article, Authors proposed a PDX (patient-derived xenograft) model to resemble high-risk and recurrent EC patients by using WES data and immunohistochemistry analysis. The ability of PDX to recapitulate key aspects of human malignancies by retaining histological and molecular markers from the patient seems to have the potential to be used in EC and to become a powerful tool for drug-testing assays and drug-response biomarker identification.

In my opinion, this article gives a valid contribution to the field.

I have the following comments to the Authors:

  • Please use the following order of sections in order to make the article more easily readable: Introduction, Materials and Methods, Results, Discussion, Conclusion
  • Methods: In order to make the study reproducible, Authors should describe in details how were selected the patients included in the study and how was selection bias excluded during this phase.
  • Discussion: Authors should include in the text a section describing strengths and limitations of this study, in order to help other researchers to provide further studies about this topic.
  • Discussion: The ESTRO/ESGO/ESP guidelines for the management of EC proposed a novel risk stratification model including molecular TCGA molecular groups to assess the prognosis of EC in association with classic, well-known, clinicopathologic prognostic factors of EC (such as myometrial invasion, histotype or lymph vascular space invasion). In this study, Authors described a PDX model to resemble high-risk and recurrent EC, which could be very useful in the future to study EC behavior, in particular if any molecular group is be related to a particular histologic factor (and so prognosis), which is a very hot topic in literature to date (PMID: 34088515; PMID: 35078650). Authors may expand discussion with potential applications this model may have in research about EC.

Author Response

We appreciate reviewer comment. Following a point-by-point reply to every reviewer´s request

- Please use the following order of sections in order to make the article more easily readable: Introduction, Materials and Methods, Results, Discussion, Conclusion

Answer: The sections order is a prerequisite from the journal. We just fit our work to the guidelines for research articles of IJMS.

- Methods: In order to make the study reproducible, Authors should describe in details how were selected the patients included in the study and how was selection bias excluded during this phase.

Answer: We agree with reviewer and we have included in Material & method section 4.1 the following information: “This was a retrospective study including patients with a final diagnosis of endometrioid EC with FIGO Stage IB or higher and histological grade 2 or 3; or with a non-endometrioid EC; all of them were women above 45 years of age. In this study, patients having at least a successful PDX model from two different areas of the primary tumor (PT), metastasis, or recurrence were selected.”

We have recruited and sequenced 20 different patients and their corresponding PDX models. However, not all PDX models or patients were successfully sequenced, and this reduced our study cohort to 13 patients.

We have modified Material and Method section 4.1 and added a sentence indicating the inclusion criteria for patients and justified the final number of patients studied, as aforementioned.  

- Discussion: Authors should include in the text a section describing strengths and limitations of this study, in order to help other researchers to provide further studies about this topic.

Answer: Following reviewer suggestion, we have now included the strengths and limitations of this study at the end of the discussion section.

- Discussion: The ESTRO/ESGO/ESP guidelines for the management of EC proposed a novel risk stratification model including molecular TCGA molecular groups to assess the prognosis of EC in association with classic, well-known, clinicopathologic prognostic factors of EC (such as myometrial invasion, histotype or lymph vascular space invasion). In this study, Authors described a PDX model to resemble high-risk and recurrent EC, which could be very useful in the future to study EC behavior, in particular if any molecular group is be related to a particular histologic factor (and so prognosis), which is a very hot topic in literature to date (PMID: 34088515; PMID: 35078650). Authors may expand discussion with potential applications this model may have in research about EC.

Answer: We appreciate reviewer point of view on such interesting topic. Due to our limited number of cases, it is not possible to establish any association between molecular landscape of PDX models or patients to clinicopathological features of patients. Actually, we did not found any significant association between histological classification and any parameter evaluated (age, histologic grade, myometrial invasion, lymph node invasion) among groups, as stated in the result section.

However, we are also interested in studying the differences between different tumor features on top of the molecular subgroup, as published in PMID: 34088515; PMID: 35078650 through a larger study using our clinical database and the large cohort of EC PDX models that we have in our group. Further research could unveil treatment response dedicated to specific subsets of patients: LVI positive vs LVI negative, etc. Furthermore, since we also generated PDX models from metastatic areas, and some of them were from lymph node tissues, it would be interesting to associate the specific subset of gene alterations found in these areas to the ability of tumor cells to migrate and invade lymph nodes. This kind of analysis is out of the scope of the present work, but certainly, we will address in the future.

We have extended discussion section highlighting the potential uses of preclinical PDX models for expanding our knowledge in EC disease. 

Round 2

Reviewer 1 Report

The response of the authors, together with the addition of citations on previously published manuscripts reporting the activity of some drugs in these models partially respond to my major question/comment.

I am satisfied with teh new version

Reviewer 2 Report

thank you for attaching the documents. I have reviewed all of them. To whom it may concern the paper is suitable for publication in the present form.